perinatal depression; perinatal anxiety; psychosocial intervention; nonspecialists; implementation science

**Corresponding author:**
Prasansa Subba;
Email: prasansa.subba@liverpool.ac.uk

# Feasibility and acceptability of community-based psychosocial interventions delivered by nonspecialists for perinatal common mental disorders: A systematic review using an implementation science framework

Prasansa Subba[1,2,3] ⓘ, Pragya Shrestha[2,4], Atif Rahman[1,5] ⓘ, Nagendra Luitel[2] ⓘ, Ahmed Waqas[1,5] ⓘ and Siham Sikander[1,5] ⓘ

[1]Department of Primary Care and Mental Health, University of Liverpool, Liverpool, UK; [2]Research Department, Transcultural Psychosocial Organization Nepal, Kathmandu, Nepal; [3]Department of Public Health, University of Copenhagen, Copenhagen, Denmark; [4]Faculty of Buddhist Studies, Lumbini Buddhist University, Lumbini, Nepal and [5]Human Development Research Foundation, Islamabad, Pakistan

## Abstract

Task sharing is endorsed as one of the strategies to address the treatment gap in common perinatal mental health conditions. There is a well-established body of evidence on the effectiveness of psychological interventions delivered by nonspecialist health workers (NSHWs); however, there is a dearth of evidence documenting factors determining the feasibility, acceptability and sustainability of integrating and implementing these interventions. This systematic review aims to synthesize the implementation outcomes and implementation process of NSHWs-delivered psychological interventions for the management of perinatal depression and anxiety using Proctor's implementation science framework outlining eight constructs: feasibility, acceptability, appropriateness, adoption, cost, fidelity, penetration and sustainability. We searched PubMed, Web of Science and Cochrane Center Register of Controlled Trials for studies published in English and between 2000 and 2022 using search terms under five broad categories: (a) "perinatal"; (b) "common mental disorders"; (c) "psychological interventions"; (d) "nonspecialist" and (e) "implementation outcomes." Secondary publications were also hand-searched for data extraction. Two authors independently reviewed abstracts and full-text articles. Data for included articles were extracted using a standard data extraction sheet. A narrative synthesis of qualitative evidence was conducted. Initial searches identified 885 articles of which full text of 128 articles were screened for eligibility, with 56 studies meeting the inclusion criteria. Out of the eight constructs of Proctor's framework, "feasibility," "acceptability," "appropriateness" and "fidelity" were the most evaluated outcomes. None of the studies reported "penetration" and very few reported "sustainability," "adoption" or "cost." None of the studies used any implementation science framework for the study evaluation. Despite the well-established evidence on the effectiveness of psychosocial interventions for perinatal depression and anxiety by NSHWs, these interventions are rarely adopted into the health system. More studies applying systems thinking are needed to explore facilitators, barriers and mechanisms for integrating interventions in the health system. Using implementation science frameworks to design, plan, execute and evaluate psychosocial interventions by NSHWs can address this gap in evidence.

## Impact statement

This review synthesizes evidence on the implementation of psychological interventions for perinatal depression and anxiety delivered by nonspecialist health workers (NSHWs). Using Proctor's framework, it highlights the successes, challenges and processes involved in these interventions offering insights for policymakers, healthcare administrators and practitioners to improve perinatal mental health programs. The review finds that NSHWs can deliver psychological intervention effectively if they are well-trained, supervised and properly incentivized. These interventions are more successful when they fit well with the local culture and integrated within the existing system. However, there is a critical gap in understanding the larger systems that affect the long-term success of these interventions. The review highlights the need for further research on how these programs can be integrated and sustained within the system.

## Introduction

Depression and anxiety are the most common perinatal (pregnancy up to 1 year postnatal) mental disorders (Waqas et al., 2022). Approximately 15% and 25% of women suffer from perinatal anxiety and depression and the burden is higher in low- and middle-income countries (LMICs) compared to high-income countries (HICs) (Nielsen-Scott et al., 2022; Mitchell et al., 2023). Perinatal mental disorders are associated with maternal suicide, poor uptake of health services, delayed social, emotional and cognitive development in infants, and marital discord (Dagher et al., 2021; Kroh and Lim, 2021; Wang et al., 2021; Stewart and Payne, 2023). Despite its debilitating effects on the woman, her infant and her social relationships, detection and treatment of perinatal depression remains a challenge (Gelaye et al., 2016). Evidence suggests that more than 80% of women with perinatal depression are out of care (Cox et al., 2016) and less than 40% intend to seek help (Daehn et al., 2022). This "treatment gap," the gap between the need and access to treatment, is more prominent in marginalized populations such as women in rural areas, from ethnic minorities, or with poor socioeconomic status ( Stirling et al., 2001; Price and Proctor, 2009; Prady et al., 2021).

Challenges pertaining to the treatment gap can be broadly categorized into demand and supply-side challenges. Lack of awareness about depression, its treatment options, treatment availability, stigma, time constraints and the practice of "wait and get it over naturally" are common barriers impeding women to seek help (Dagher et al., 2021; Iturralde et al., 2021). Further, poor investment in mental health, scarcity of skilled and trained human resources, ill-equipped health facilities, stigma and lack of health professionals' awareness contribute to the expanding treatment gap (Lasater et al., 2017; Dagher et al., 2021). The World Health Organization (WHO) (2021) reports that 50% of the world's population lives in a place where there is less than one psychiatrist for 100,000 population. The WHO advocates for a task-sharing approach whereby expert knowledge and skills are transferred to nonspecialist health workers (NSHWs) (WHO, 2016). Psychological interventions are a first-line treatment recommended for perinatal depression. There is a well-established evidence base that shows psychological interventions delivered by NSHWs are effective both at preventing (Prina et al., 2023) and treating perinatal depression (Singla, Lawson, et al., 2021), but they do not adequately address the questions of "how" interventions can be successfully integrated and adopted across diverse contexts.

A review by Munodawafa (2018) discusses the context and mechanisms of successful implementation of interventions for perinatal depression. Additional evidence on intervention content for perinatal depression and its delivery in LMICs (Chowdhary et al., 2014) and HICs (Singla, Lawson, et al., 2021) also exists; however, a combined global evidence on the evaluation of these interventions using implementation science constructs is still lacking. As NSHWs continue to be an important cadre for delivering services for perinatal depression, it is important to understand how best they can be mobilized. Proctor et al (2011) have proposed eight constructs for documenting implementation outcomes, namely: acceptability, adoption, appropriateness, feasibility, fidelity, implementation cost, penetration and sustainability. The current systematic review thus aims to synthesize evidence on the implementation process of NSHW-delivered psychosocial interventions for the management of perinatal depression and anxiety, as well as implementation outcomes based on the Proctor's framework (Proctor et al., 2011). The findings from this review will be valuable to policymakers, practitioners and academics working on task-sharing interventions to address perinatal mental health concerns.

## Methods

The protocol for this review was registered in the National Institute for Health Research with the PROSPERO registration No. CRD42022306566 on March 10, 2022. This systematic review followed the Preferred Reporting Items for Systematic Review guidelines for reporting (Page et al., 2021).

### Search strategy

The first (PS1) and second author (PS2) performed the search in three databases: PubMed, Web of Science and Cochrane Center Register of Controlled Trials. Search strategies were developed for each database using terms for five broad responses: "perinatal," "common mental disorders," "psychological interventions," "nonspecialist" and "implementation" and filtered by date (1 January 2000 and 1 January 2022) (see Table 1). Full search strategy tailored to each database can be found in Supplementary File 1.

### Screening

Two reviewers, PS1 and PS2, independently screened the titles and abstracts of studies identified through the database search. The full text of each article was then reviewed for eligibility.

### Data extraction

Data relevant to this review were extracted from selected papers into a Microsoft Excel spreadsheet. In the first step, the selected papers were evenly distributed between the two reviewers, PS1 and PS2. Each reviewer independently extracted relevant information and categorized it under the following headings: author, year, setting, design, intervention details, delivery agents, training, supervision, feasibility, acceptability, fidelity, barriers, facilitators, appropriateness, adoption, implementation cost, penetration and sustainability. In the subsequent step, PS1 and PS2 cross-reviewed each other's data extraction tables to verify their accuracy and completeness. Any disagreements between the reviewers were discussed with AW.

### Quality assessment

PS1 appraised all studies and discussed any confusion with AW. The Critical Appraisal Skills Program (CASP) checklist was used for qualitative studies (Critical Appraisal Skills Programme, 2018). The CASP checklist examines methods, study design, positionality, data collection and analysis procedures where studies are rated "yes," "no," "insufficient" or "not applicable." For quantitative and mixed-method studies, an assessment tool designed and used by Liu et al. (2019) was used. Studies were rated as "yes," "no," "partially," "unclear" or "not applicable" under domains such as planning, design and conduct and reporting stages. Both the checklists do not use any quantitative scoring system.

### Data synthesis

Narrative synthesis was used to synthesize data on intervention implementation (Popay et al., 2006). We initiated a preliminary

**Table 1.** Search strategy adapted for *PubMed* database

| | Concept | Keywords |
|---|---|---|
| Population | Perinatal | (antenatal[Title/Abstract] OR antepartum[Title/Abstract] OR pregnan*[Title/Abstract] OR postpartum[Title/Abstract] OR postnatal[Title/Abstract] OR maternal[Title/Abstract] OR perinatal[Title/Abstract] OR peripartum[Title/Abstract] OR postpartum period[MeSH Terms]) |
| Condition | Common mental disorders | ("depressive disorder*"[Title/Abstract] OR "depression"[Title/Abstract] OR "depress*"[Title/Abstract] OR "anxiety"[Title/Abstract] OR "anxiety disorder*"[Title/Abstract] OR "anxi*"[Title/Abstract] OR "common mental disorder*"[Title/Abstract] OR "common mental health problem"[Title/Abstract] OR "anxiety disorder" [MeSH Terms] OR "depressive disorder"[MeSH Terms] OR "depression, postnatal"[MeSH Terms]) |
| Intervention | Psychological interventions | ("psychosocial counseling"[Title/Abstract] OR psychoeducat*[Title/Abstract] OR "non-pharmacological"[Title/Abstract] OR psychotherapy[Title/Abstract] OR "psychological therapy"[Title/Abstract] OR "group therapy"[Title/Abstract] OR "group counseling"[Title/Abstract] OR "group counseling"[Title/Abstract] OR "individual counseling"[Title/Abstract] OR "individual counseling"[Title/Abstract] OR "group session*"[Title/Abstract] OR "non-directive counseling"[Title/Abstract] OR "non-directive counseling"[Title/Abstract] OR "comprehensive psychosocial intervention*" [Title/Abstract] OR "multifaceted intervention*"[Title/Abstract] OR "integrated mental health service*"[Title/Abstract] OR "multicomponent intervention*"[Title/Abstract] OR "multidimensional intervention*"[Title/Abstract] OR "holistic intervention*" [Title/Abstract] OR "community based intervention*"[Title/Abstract] OR "cognitive behavioral therapy"[Title/Abstract] OR "dialectical behavior therapy"[Title/Abstract] OR "interpersonal therapy"[Title/Abstract] OR "relational therapy"[Title/Abstract] OR "talk therapy"[Title/Abstract] OR "psychosocial intervention"[MeSH Terms] OR "cognitive behavioral therapy"[MeSH Terms] OR "Dialectical Behavior Therapy"[MeSH Terms] OR "Interpersonal Psychotherapy"[MeSH Terms]) |
| | Nonspecialist | ("nonspecialist*"[Title/Abstract] OR "task shar*"[Title/Abstract] OR "task shift*"[Title/Abstract] OR "community health worker*"[Title/Abstract] OR "lay health worker*"[Title/Abstract] OR "peer volunteer*"[Title/Abstract] OR "community volunteer*"[Title/Abstract] OR "health worker*"[Title/Abstract] OR "barefoot doctor*"[Title/Abstract] OR "psychosocial worker*"[Title/Abstract] OR "psychosocial counselor*"[Title/Abstract] OR "specially trained"[Title/Abstract] OR "nurse*"[Title/Abstract] OR "village health worker*"[Title/Abstract] OR "community health aide"[Title/Abstract] OR "community health workers"[MeSH Terms]) |
| Outcomes | Implementation outcomes | ("clinical competence"[MeSH Terms] OR "program evaluation"[MeSH Terms] OR "feasibility"[Title/Abstract] OR "acceptability"[Title/Abstract] OR "evaluation"[Title/Abstract] OR "program evaluation"[Title/Abstract] OR "program"[Title/Abstract] OR "fidelity"[Title/Abstract] OR "implementation"[Title/Abstract] OR "practice"[Title/Abstract] OR "reach"[Title/Abstract] OR "penetration[Title/Abstract] OR "train*"[Title/Abstract] OR "clinical mentor*"[Title/Abstract] OR "clinical supervision"[Title/Abstract] OR "clinical competenc*"[Title/Abstract] OR "competenc*"[Title/Abstract] OR "attitude*"[Title/Abstract] OR "perception*"[Title/Abstract] OR "view*"[Title/Abstract] OR "behavior*"[Title/Abstract] OR "sustainab*"[Title/Abstract] OR "facilitator*"[Title/Abstract] OR "barrier*"[Title/Abstract]) |

synthesis of the data as per the eight constructs of Proctor's framework (Proctor et al., 2011) and explored relationships based on the intervention characteristics and study design. Where information was missing under certain outcomes, additional articles from the same studies were examined. Six additional articles (Glavin et al., 2010a; Segre et al., 2010; Segre et al., 2015; Lund et al., 2020; Davies et al., 2022; Yator et al., 2022) related to the included studies (Glavin et al., 2010b; Brock et al., 2017; Boisits et al., 2021; Yator et al., 2021) were reviewed for additional data. This review focused on the implementation process outcomes; hence, a meta-analysis was not conducted.

## Results

### Study selection

A total of 885 studies were retrieved, with 117 duplicates. After screening the titles/abstracts, 128 studies were reviewed in full and 56 met the inclusion criteria. Reasons for exclusion included inappropriate interventions, study design, specialist-delivered, hospital settings or language other than English (see Figure 1).

### Quality assessment

Altogether, 15 qualitative studies were assessed based on the CASP checklist. While most qualitative studies provided clear objectives, methodology and findings, there was inadequate reporting on the researcher's positionality (60%), the value of the research (46.66%) and ethical issues (40%). A few studies focused on process

documentation (Eappen et al., 2018; Yator et al., 2021), adaptation and development of interventions (Zayas et al., 2004); therefore, the study methods and data analysis were not applicable.

Overall, studies employing quantitative and mixed methods (n = 41) had adequately described their purpose (n = 39), interventions (n = 38) and study methods (n = 34). Implementation outcomes as per Proctor's framework were reported partially by 36 studies with most examining feasibility, training and supervision outcomes. The included trials and pilot studies poorly reported on study team (n = 11), transparency of data analysis (n = 9) and protocol registration (n = 13) (see Supplementary File 2).

### Description of studies

Twenty-four studies were published in HICs, followed by LMICs (n = 23) and upper middle-income countries (n = 9). Most studies were quantitative (n = 30), followed by qualitative (n = 15) and mixed methods (n = 10). Studies ranged from intervention development to implementation and effectiveness testing. One study reported it as a prevention intervention (Zayas et al., 2004), but the reference article (Miranda and Muñoz, 1994) clarified that it targeted mild depression, justifying its inclusion. Details are given in Table 2.

### Implementation process

#### Intervention details
Most interventions targeted postnatal depression (n = 26), followed by perinatal (n = 22), antenatal (n = 5) and maternal depression (n = 3). Anxiety (Prendergast and Austin, 2001; Boisits et al., 2021),

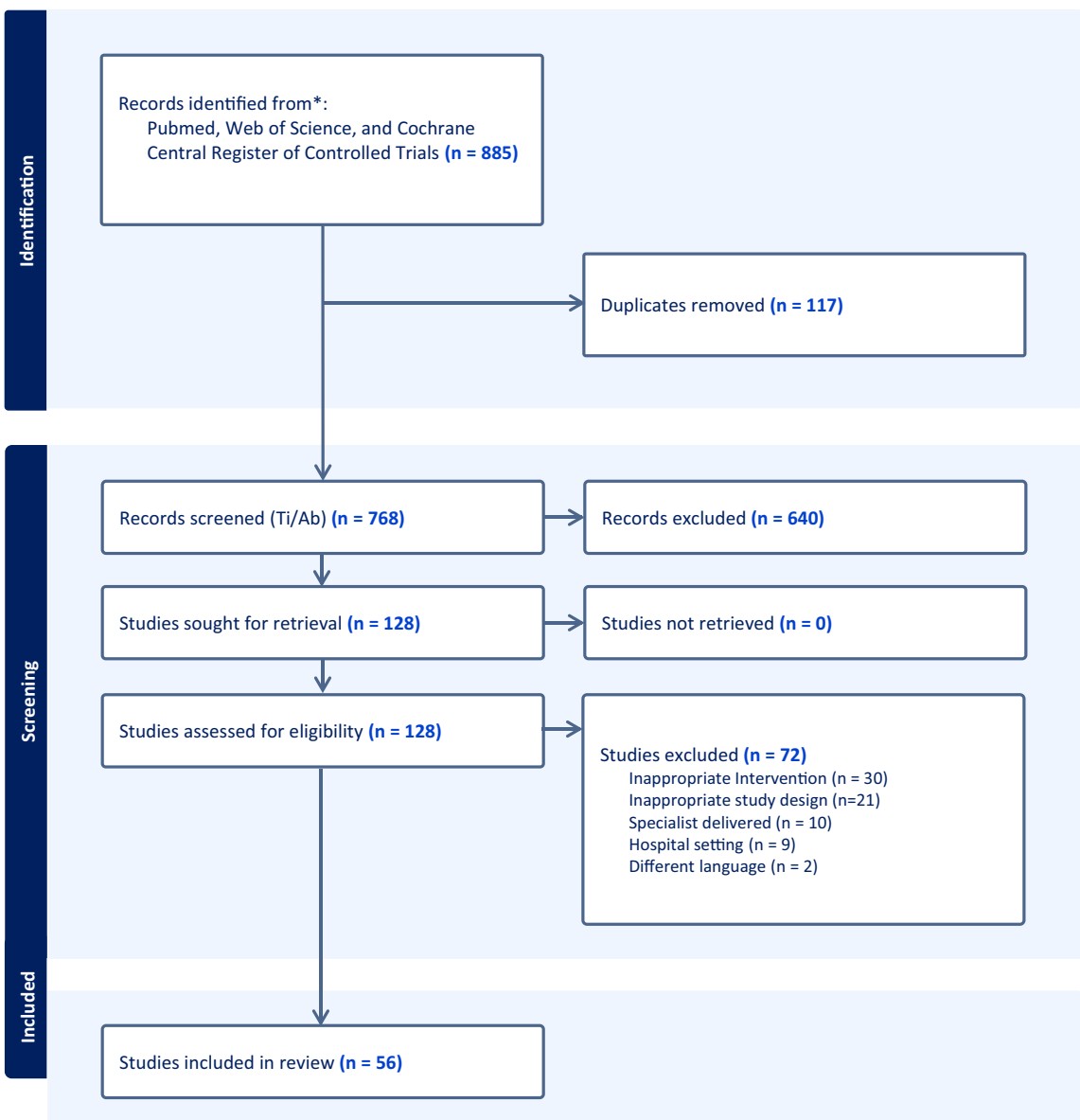

**Figure 1.** PRISMA flow diagram.

parenting (Sawyer et al., 2019; Husain et al., 2021), infant development (Zayas et al., 2004) and mother–infant relationship (Horowitz et al., 2013; Atif, Bibi, et al., 2019) were also addressed in some interventions. Cognitive behavioral therapy was the most widely used approach (n = 28), followed by problem-solving therapy (n = 5) and interpersonal therapy (IPT) (n = 4). Telephone-based interventions provided peer support (n = 4), psychoeducation (n = 1) or IPT sessions (n = 2). Most interventions (six studies missing information) were delivered in-person at home, health facilities or community centers (n = 40), followed by remote (n = 8) and hybrid sessions (n = 2). Session lasted between 15 min and 2 h, with individual sessions generally being shorter. Refer to Table 2 for further details.

*Delivery agents and their characteristics*
Nurses/midwives (n = 24) were the most common cadres, followed by peers (n = 15), community health workers (CHWs) (n = 14), school teachers/local priests (n = 1) (Notiar et al., 2021) and graduate students

(n = 1) (Zayas et al., 2004). One study on intervention development did not mention the occupations of the NSHWs (Ng'oma et al., 2019). Nurses/midwives typically held diplomas or master's degrees or had extensive nursing experience, but no mental health training. Peers in LMICs were local married women sharing similar culture and socio-economic status (Atif, Bibi, et al., 2019; Atif, Nisar, et al., 2019; Fuhr et al., 2019; Rahman et al., 2021). Peers in HICs were matched by lived experience of perinatal depression (Dennis, 2003, 2010, 2013, 2014; Letourneau et al., 2011; Amani et al., 2021). In Zimbabwe, health facilities providing prevention services for mother-to-child HIV transmission trained and mobilized HIV-infected women as peer counselors (Chibanda et al., 2014). CHWs were often local females, with at least secondary education and 2.5 years of work experience in maternal and child health programs.

*Training*
Forty-three studies reported details on the training for the NSHWs, while information from two studies (Brock et al., 2017;

**Table 2.** Study description and key characteristics of intervention

| Author year | Methods | | | Intervention details | | | | | | Delivery agents |
| | Country | Study design | Aims | Theoretical underpinning | Intervention target | Mode of delivery | Type | Number of sessions | Duration | Type of NSHW |
| --- | --- | --- | --- | --- | --- | --- | --- | --- | --- | --- |
| VanLieshout 2020 | Canada | Quantitative | To test feasibility and acceptability of training the nurses to deliver intervention | CBT | Postnatal depression | In-person | Group | Nine | 120 min | Nurses |
| Posmontier 2016 | USA | Quantitative | To test feasibility and acceptability of the intervention | IPT | Postnatal depression | Telephone | Individual | Eight | 50 min | Nurses |
| Leocata 2021 | India | Qualitative | Ethnographic study to explore fidelity using manuals to guide sessions | CBT | Perinatal depression | In-person | Individual | 15 | N/A | Peers |
| Boisits 2021 | South Africa | Mixed methods | To describe intervention development process | CBT and problem-solving | Perinatal depression and anxiety | In-person | Individual | Three | 45 min | Community Health Worker |
| Fuhr 2019 | India | Quantitative | To assess the effectiveness and cost-effectiveness of THP | CBT | Perinatal depression | In-person | Individual | 6–14 | 30–45 min | Community Health Worker |
| Munodawafa 2017 | South Africa | Qualitative | To explore barriers and facilitators of task-sharing intervention | CBT, IPT and problems solving | Antenatal depression | In-person | Individual | Six to eight | N/A | Community Health Worker |
| Glavin 2010b | Norway | Quantitative | To examine effectiveness of the intervention | Non-directive counseling | Postnatal depression | In-person | Individual | Individualized | 30 min | Nurses |
| Singla 2020 | India | Mixed methods | To explore barriers and facilitators of peer supervision and validate tool | CBT | Perinatal depression | In-person | Individual | 14 | 45 min | Peers |
| Dennis 2013 | Canada | Mixed methods | To explore service providers' experience delivering the intervention | Emotional and peer support | Postnatal depression | Telephone | Individual | Minimum four | Individualized | Peers |
| Ransing 2021 | India | Qualitative | To describe intervention development process | Psychoeducation, relaxation, health promotion | Perinatal depression | In-person | Individual | Three | 15–20 min | Community Health Worker |
| Craig 2005 | Australia | Quantitative | To examine effectiveness of the intervention | CBT | Postnatal depression | In-person | Group | Nine | 120 min | Community Health Worker |
| Rahman 2007 | Pakistan | Qualitative | To explore barriers and facilitators developing and delivering intervention | CBT | Perinatal depression | In-person | Individual | 16 | 45 min | Community Health Worker |
| Morrell 2009 | England | Quantitative | To examine effectiveness of the intervention | CBT or non-directive counseling | Postnatal depression | In-person | Individual | Eight | 60 min | Midwives/nurse |
| Eappen 2018 | Peru | Qualitative | To describe preparatory steps before implementation | CBT | Perinatal depression | In-person | Individual | N/A | N/A | Community health worker |

*(Continued)*

**Table 2.** (*Continued*)

| Author year | Country | Study design | Aims | Theoretical underpinning | Intervention target | Mode of delivery | Type | Number of sessions | Duration | Type of NSHW |
|---|---|---|---|---|---|---|---|---|---|---|
| | | **Methods** | | | | **Intervention details** | | | | **Delivery agents** |
| Dennis 2014 | Canada | Mixed methods | To describe development and implementation process | Emotional and peer support | Postnatal depression | Telephone | Individual | Minimum four | Individualized | Peers |
| Husain 2021 | Pakistan | Quantitative | To test efficacy of the intervention | Problem solving; psychoeducation and parenting | Maternal depression and parenting | In-person | Group | 10 | 60–90 min | Community health worker |
| Letourneau 2011 | Canada | Quantitative | To examine effectiveness of the intervention | Peer support | Postnatal depression | In person or through telephone | Individual | 12 | 20 min | Peers |
| Özkan 2020 | Turkey | Quantitative | To examine effectiveness of the intervention | Exercise | Postnatal depression | In-person | Mixed | Regularly for 4 weeks | N/A | Midwives/nurses |
| Slade 2010 | England | Qualitative | To explore experience of postnatal women engaging in intervention | CBT or person-centered intervention | Postnatal depression | In-person | Individual | Up to eight | 60 min | Midwives/nurse |
| Singla 2021 | India and Pakistan | Quantitative | To explore mediators influencing effectiveness of intervention | CBT | Perinatal depression | In-person | Individual (India); Group (Pakistan) | 14 | Up to 45 min | Peers |
| Nyatsanza 2016 | South Africa | Qualitative | To describe development process of manual | CBT, problem-solving | Perinatal depression | In-person | Individual | Six | N/A | Midwives |
| Nakku 2021 | Uganda | Quantitative | To examine effectiveness of the intervention | Problem Solving | Perinatal depression | In-person | Group | Minimum four | 90–120 min | Midwives |
| Roman 2009 | USA | Quantitative | To examine effectiveness of the intervention | Problem-solving | Perinatal depression | In person or through telephone | Individual | Up to 18 | N/A | Nurses |
| Dennis 2003 | Canada | Quantitative | To explore feasibility and preliminary efficacy of intervention | Peer support | Postnatal depression | Telephone | Individual | N/A | N/A | Peers |
| Nisar 2020 | China | Mixed methods | To adapt intervention and explore acceptability | CBT | Perinatal depression | N/A | N/A | Seven | N/A | Nurses |
| Russell 2020 | USA | Qualitative | To explore users' experience engaged in intervention | Motivational interviewing and mindful exercise | Antenatal depression | In-person | Group | 12 | 75 min | Nurses |
| Zayas 2004 | USA | Qualitative | To describe lessons learned during intervention implementation | CBT | Perinatal depression | In-person | Individual | Eight | N/A | Graduate students |
| Ross 2013 | Thailand | Mixed methods | To examine effectiveness of the intervention | Emotional and informational support | Antenatal depression | Telephone | Individual | N/A | 15–30 min | Nurses |
| Tezel 2006 | Turkey | Quantitative | To examine the impact of the intervention | Problem solving | Postnatal depression | In-person | Individual | N/A | N/A | Nurses |

(*Continued*)

| Author year | Country | Study design | Aims | Theoretical underpinning | Intervention target | Mode of delivery | Type | Number of sessions | Duration | Type of NSHW |
|---|---|---|---|---|---|---|---|---|---|---|
| | | Methods | | | Intervention details | | | | | Delivery agents |
| Leocata 2021 | India | Qualitative | To explore service providers' experience delivering the intervention | CBT | Perinatal depression | In-person | Individual | N/A | N/A | Peers |
| Sawyer 2019 | Australia | Quantitative | To examine effectiveness of the intervention | Maternal self-efficacy and social support mechanism | Postnatal depression and parenting problems | Virtual App based | Group | N/A | N/A | Nurses |
| Yator 2021 | Kenya | Qualitative | To describe implementation process and acceptability of intervention | IPT | Postnatal depression | In-person | Group | 8 | N/A | Community health worker |
| Tryphonopoulos 2020 | Canada | Quantitative | To test efficacy of the intervention | Barnard Model Maternal–infant interaction | Postnatal depression and maternal infant interaction | In-person | Mother–infant dyad | Three | 60–90 min | Nurses |
| Chibanda 2014 | Zimbabwe | Quantitative | To test efficacy of the intervention | Problem solving | Postnatal depression | In-person | Group | 12 | 60 min | Peers |
| Nisar 2022 | China | Quantitative | To examine effectiveness of electronic vs. face-to-face training | CBT | Perinatal depression | N/A | N/A | N/A | N/A | Nurses |
| Brock 2017 | USA | Quantitative | To assess the sustainability of treatment effect | Problem solving | Antenatal depression | In-person | Individual | Six | 30–50 min | Midwives/nurse |
| Notiar 2021 | Kenya | Mixed methods | To test the acceptability and feasibility of the intervention | CBT | Maternal depression | In-person | Group | 12 | 60–90 min | School teacher/priest |
| Layton 2020 | Canada | Qualitative | To explore NSWHs' experience delivering the intervention | CBT | Postnatal depression | In-person | Group | Nine | 120 min | Nurses |
| Carter 2020 | UK | Qualitative | To test feasibility and acceptability of the intervention | Peer support | Antenatal depression | In-person | Individual | Six | 60 min | Peers |
| Sikander 2019 | Pakistan | Quantitative | To examine effectiveness and cost-effectiveness of intervention | CBT | Perinatal depression | In-person | Mixed | 14 | N/A | Peers |
| Horowitz 2013 | United States | Quantitative | To test efficacy of the intervention | CBT | Postnatal depression and mother and child bond | In-person | Individual | Two | 30–40 min | Nurses |
| Appleby 2003 | UK | Quantitative | To test acceptability of training the non-specialists | CBT | Postnatal depression | N/A | N/A | N/A | N/A | Midwives/nurse |
| Rahman 2008 | Pakistan | Quantitative | To test effectiveness of the intervention | CBT | Perinatal depression and nutrition | In-person | Individual | 16 | N/A | Community health worker |

(*Continued*)

**Table 2.** (*Continued*)

| Author year | Country | Study design | Aims | Theoretical underpinning | Intervention target | Mode of delivery | Type | Number of sessions | Duration | Type of NSHW |
|---|---|---|---|---|---|---|---|---|---|---|
| | | Methods | | | Intervention details | | | | | Delivery agents |
| Gureje 2019 | Nigeria | Quantitative | To test effectiveness of the intervention | Problem solving | Perinatal depression | In-person | Individual | 8–16+ | N/A | Community health worker |
| Prendergast 2001 | Australia | Quantitative | To test feasibility of training the non-specialists and its impact | CBT | Postnatal depression and anxiety | In-person | Individual | Six | 60 min | Nurses |
| Dennis 2020 | Canada | Quantitative | To test effectiveness of the intervention | IPT | Postnatal depression | Telephone | Individual | 12 | 60 min | Nurses |
| Yator 2021 | Kenya | Quantitative | To test feasibility and acceptability of the intervention | IPT | Postnatal depression | In-person | Group | N/A | N/A | Community health worker |
| Amani 2021 | Canada | Quantitative | To test effectiveness of the intervention | CBT | Postnatal depression | In-person | Group | Nine | 120 min | Peers |
| Rahman 2019 | Pakistan | Quantitative | To examine effectiveness of electronic vs. face-to-face training | CBT | Perinatal depression | N/A | N/A | N/A | N/A | Community health worker |
| Atif 2019 | Pakistan | Mixed methods | To describes the process of training and supervision of NSHW | CBT | Perinatal depression | N/A | N/A | N/A | N/A | Community health worker |
| Tomlinson 2020 | Afghanistan | Mixed methods | To test feasibility of training NSHWs for intervention delivery | CBT | Postnatal depression | In-person | Individual | N/A | N/A | Midwives |
| Atif 2019 | Pakistan | Mixed methods | To describe NSHWs experience after project completion | CBT | Maternal depression | In-person | Group | 18+ | N/A | Peers |
| Ng'oma 2019 | Malawi | Qualitative | To describe intervention development process | N/A | Perinatal depression | N/A | N/A | N/A | N/A | N/A |
| Atif 2016 | Pakistan | Qualitative | To explore NSHWs' perceived feasibility and acceptability of intervention delivery prior to implementation | CBT | Perinatal depression | In-person | Group | N/A | N/A | Peers |
| Dennis 2010 | Canada | Quantitative | To describe experiences of women engaged in intervention | Peer support | Postnatal depression | Telephone | Individual | Individualized | Individualized | Peers |
| VanLieshout 2022 | Canada | Quantitative | To examine effectiveness of group intervention | CBT | Perinatal depression | In-person | Group | Nine | 120 min | Nurses |

Abbreviations: CBT, cognitive behavioral therapy; IPT, interpersonal therapy; N/A, not available.

**Table 3.** Implementation outcomes as per the Proctor's Framework

| Study ID | 1. Feasibility | 2. Acceptability | 3. Appropriateness | 4. Adoption | 5. Fidelity | 6. Cost | 7. Penetration | 8. Sustainability |
|---|---|---|---|---|---|---|---|---|
| VanLieshout 2020 | Screening tool- Edinburgh Postnatal Depression Scale (EPDS) 7/12 were enrolled; 71.43% treatment adherence 100% response rate Service providers training completion rate– 100% | Childcare facility facilitated to attend sessions | Group sessions helped in normalization and strengthened bonding | N/A | N/A | N/A | N/A | N/A |
| Posmontier 2016 | Screening tool- Hamilton Rating Scale for Depression (HAM-D) | N/A | Phone based sessions reduced stigma, improved comfort and saved transportation hassle | N/A | Method: Audio recording Raters: IPT Supervisors, experts Result: Not mentioned (methods only) | N/A | N/A | N/A |
| Leocata 2021 | N/A | N/A | N/A | N/A | Method: Therapy Quality Scale Raters: N/A Result: N/A | N/A | N/A | N/A |
| Boisits 2021 | Screening tool- EPDS Recruitment– 19.4% Follow up rate at 3 months– 93.5% for intervention and 80.4% for control Intervention completion– 53.3% | Metaphors, brief sessions eased service delivery. **Negative experience-NSHWs Inconsideration, manual reading, giving high hopes Positive experience- Psychoeducation and normalization, space to share, connection with NSHW.** | Stepwise approach was helpful. Need identified for more time, training and supervision. | Brief intervention (3 sessions) was easy to adopt by NSHWs | N/A | N/A | N/A | N/A |
| Fuhr 2019 | Screening tool- Patient Health Questionnaire (PHQ– 9) 84·1% consented to participate; Treatment completion rate– 71·7%. 11·5% were discontinued and referred to specialist at 6 months | N/A | **Family member engagement, and need to address incentives** | N/A | Method: Audio recording rated on Therapy Quality Scale Raters: N/A Result: N/A | Mean cost: USD 1.36 (12% attributed to incentives) | N/A | N/A |
| Munodawafa 2017 | 47.8% completed six sessions. 27.2% dropped out 26.7% women did not complete all 6 sessions. | **Secondary article- Lack of empathy, reading from the manual, giving high hopes were barriers while normalization, strong bond, hope for change, having a space to share problems were facilitators.** | N/A | Challenges negotiating the clinic environment. Not welcomed by the health facility staff stood as a barrier | Method: Audio recording Raters: Supervisor Result: 62.8% fidelity rating meaning moderate to good adherence to the manual | N/A | N/A | N/A |

**Table 3.** (*Continued*)

| Study ID | 1. Feasibility | 2. Acceptability | 3. Appropriateness | 4. Adoption | 5. Fidelity | 6. Cost | 7. Penetration | 8. Sustainability |
|---|---|---|---|---|---|---|---|---|
| Glavin 2010b | Screening tool- EPDS **Response rates at t2 (3 months)– 85.5%, at t3 (6 months)– 66.3%, t4 (12 months postpartum)– 54.2%** | **Training elevated their confidence to support. Screening was easy and made it feasible to integrate in daily lives.** | N/A | N/A | N/A | N/A | N/A | N/A |
| Singla 2020 | N/A | N/A | N/A | Lack of compensation; lack of time, other commitments impeded implementation | Method: Audio recording (Rating tool- Therapy Quality Scale) Raters: Peers Result: N/A | N/A | N/A | N/A |
| Dennis 2013 | N/A | Motivators- personal growth, knowledge and social network expansion, altruism, | N/A | N/A | N/A | N/A | N/A | N/A |
| Ransing 2021 | Nurses were able to screen without difficulty post-training. Nearly 50% suggested 2–3 sessions to reduce attrition | **Secondary article – 83% mothers found the intervention useful and that it contributed to their learning** | A quarter midwives expressed extra workload and suggested monetary incentivization for motivation | N/A | N/A | N/A | N/A | N/A |
| Craig 2005 | Screening Tool- Hospital Anxiety and Depression Scale (HADS) Response rates– 14/16 completed pre-post data at 6 weeks follow up– 7 (50%); 6 months follow up – 7 (50%) | N/A | N/A | N/A | N/A | N/A | N/A | N/A |
| Rahman 2007 | N/A | 48% satisfied with the intervention | NSHWs found training relevant and easy to understand | N/A | N/A | N/A | N/A | N/A |
| Morrell 2009 | Screening tool- EPDS 53% with a live baby consented to take part. | N/A | N/A | N/A | N/A | N/A | N/A | N/A |
| Eappen 2018 | Screening tool- Self-Reporting Questionnaire (SRQ–18) | N/A | N/A | Longer intervention difficult to adopt; difficult geography hindered accessibility | N/A | N/A | N/A | N/A |
| Dennis 2014 | Screening tool- EPDS Peer volunteers supported 2 mothers | Easy to maintain activity log. NSHWs felt supported and were willing to engage. Majority of mothers were satisfied, felt peer was helpful | N/A | N/A | Method: Intervention Protocol Raters: N/A Result: N/A | N/A | N/A | N/A |

(*Continued*)

| Study ID | 1. Feasibility | 2. Acceptability | 3. Appropriateness | 4. Adoption | 5. Fidelity | 6. Cost | 7. Penetration | 8. Sustainability |
|---|---|---|---|---|---|---|---|---|
| Husain 2021 | Screening tool- PHQ–9 Response rates above 90% at 3 timepoints | N/A | N/A | N/A | Method: Observation Raters: Independent senior researchers Result: N/A | N/A | N/A | N/A |
| Letourneau 2011 | Screening tool- EPDS Recruitment – Only 65 could be recruited Hospitalizations, relocation, lack of time and loss of contact led to incomplete data | N/A | N/A | N/A | N/A | N/A | N/A | N/A |
| Özkan 2020 | Screening tool- EPDS | N/A | N/A | N/A | N/A | N/A | N/A | N/A |
| Slade 2010 | Screening tool- EPDS Recruitment– 6/39 declined; 3 did not respond | Barriers: sessions as informal, intervention not tailored. Facilitators- space to discuss changes | N/A | N/A | Method: Audio recording Raters: Not mentioned Result: Good adherence | N/A | N/A | N/A |
| Singla 2021 | Screening tool- PHQ–9 | N/A | N/A | N/A | N/A | N/A | N/A | N/A |
| Nyatsanza 2016 | Screening tool – EPDS | Perceived acceptability of counseling intervention by both service users and delivery agent but 33% were fearful to share problems. | Local female counselors, monthly individual sessions perceived appropriate. 50% reported work burden. | N/A | Method: Observation during supervision and feedback Raters: N/A Result: N/A | N/A | N/A | N/A |
| Nakku 2021 | Screening tool- PHQ–9 20/30 midwives completed participation. Drop-out due to transfer and heavy workloads. 98.3% consent rate. 90.2% response rate | N/A | N/A | N/A | N/A | N/A | N/A | N/A |
| Roman 2009 | Screening tool- Center for Epidemiological Studies Depression (CES-D) NSHWs able to engage high risk women. 82% participated in the intervention (18% miscarriage, moved out of the county or withdrew). | N/A | N/A | N/A | Method: Activity log (Clinical records) Raters: Senior team member (nurse) Result: N/A | N/A | N/A | N/A |

*Cambridge Prisms: Global Mental Health*

**Table 3.** (*Continued*)

| Study ID | 1. Feasibility | 2. Acceptability | 3. Appropriateness | 4. Adoption | 5. Fidelity | 6. Cost | 7. Penetration | 8. Sustainability |
|---|---|---|---|---|---|---|---|---|
| Dennis 2003 | Screening tool- EPDS 67% consented participation. 80% response rate Average login– 5 or more actual connections and 5 attempted connections for most mothers | NSHWs were satisfied with the helping experience Mothers were satisfied, felt that they received emotional, informational and appraisal support, | Perceived NSHW as trustworthy and were willing to engage in the future. | N/A | Method: Activity logs used Raters: N/A Result: N/A | N/A | N/A | N/A |
| Nisar 2020 | Screening tool- PHQ–9 96 out of 100 students (96%) completed the assessment. | THP sessions helpful, acceptable, appropriate and meaningful Concepts clear and understandable and culturally relevant | Nurses identified as most appropriate cadre for intervention delivery | N/A | N/A | N/A | N/A | N/A |
| Russell 2020 | Screening tool- PHQ–9 **82% enrollment rate 66% completed 12 week intervention; Drop out reasons: no longer pregnant, move out of area, side effects; 9 were lost to follow up** | Group intervention provided "safe space," built "sisterhood bond," "shared experiences with other participants," helped in "normalization," learned coping mechanisms. | N/A | N/A | N/A | N/A | N/A | N/A |
| Zayas 2004 | Screening tool- Beck's Depression Inventory (BDI) 38% attrition. Retention was a challenge in studies of urban, minority poor populations. | N/A | NSHWs preferred health facility over home visits for service delivery citing safety concerns. | Logistical challenges: frequent turnover of NSHWs were noted challenges | Method: N/A Raters: N/A Result: N/A Mention of challenges not assessing fidelity | N/A | N/A | N/A |
| Ross 2013 | Screening tool- CES-D | NSHWs availability, quality of relationship, emotional support led to higher acceptability. | Telephone based support although helpful concerns of confidentiality | N/A | N/A | N/A | N/A | N/A |
| Tezel 2006 | Two steps screening using EPDS and BDI **4/70 declined participation, 3 moved to other district, 1 denied participation in the study by her spouse. Recruitment rate– 88.57%** | N/A | N/A | N/A | N/A | N/A | N/A | N/A |
| Leocata 2021 | **NSHWs retention– 77.27% Average session attendance– 7.7/10; India- total sessions 2–4 Barriers to recruitment-** | NSHW motivation- Personal development; Easy and helpful content; **adequate t**raining **and supervision,** elevated **social status, their chances of** | **Matching characteristics facilitated discussion better. Volunteerism was** | N/A | N/A | N/A | N/A | Concerns of NSHWs and service recipients on abrupt end of the program, |

(*Continued*)

*Cambridge Prisms: Global Mental Health*

| Study ID | 1. Feasibility | 2. Acceptability | 3. Appropriateness | 4. Adoption | 5. Fidelity | 6. Cost | 7. Penetration | 8. Sustainability |
|---|---|---|---|---|---|---|---|---|
| | **Refusal, migration Barriers to retention- uncontactable, withdrawal, drop out, visit to maternal home** | **employment; improvements in women and their bonding with child** | **acceptable in Pakistan was incentives expected in India.** | | | | | which could not continue beyond the study's timeframe. |
| Sawyer 2019 | Screening tool- EPDS Uncontactable (mobile population) barrier to follow-up- App login rate– 50–60% Higher in week 1 and eventually lessened | Chatpage mostly used, moodtracker least used function. App perceived informative and resourceful, | N/A | N/A | N/A | N/A | N/A | N/A |
| Yator 2021 | Screening tool- EPDS **25/32 consented to participate. Flexibility to mobilize NSHWs for follow up** | **Improved emotional and behavioral wellbeing, communication, empowerment, interpersonal skills, anger management, instilled hope.** | **NSHWs mediation role appreciated, Adequate training and supervision. NSHWs able to integrate in daily work. Ripple effect of intervention** | Lack of remuneration for NSHW affected the consistent delivery of regular sessions | Method: Audio recording/ Interpersonal inventory rating scale Raters: N/A Result: Younger NSHWs had a better understanding | N/A | N/A | N/A |
| Tryphonopoulos 2020 | Screening tool- EPDS 63% able to recruit from potential participant referrals received from community recruitment partners | N/A | N/A | N/A | N/A | N/A | N/A | N/A |
| Chibanda 2014 | Screening tool- EPDS 95% attendance rate; 85% treatment completion. 15% women were lost to follow-up with more from pharmacotherapy group. | N/A | N/A | N/A | N/A | N/A | N/A | N/A |
| Nisar 2022 | 96% post-training assessment completion rate. | E-training more acceptable, useful for learning, training | N/A | N/A | N/A | N/A | N/A | N/A |
| Brock 2017 | Screening tool- Hamilton Rating Scale for Depression (HRSD) **52.8% excluded or denied services Reasons for exclusion- language barriers, not wanting to engage in the study 6/66 did not complete assessment; Reasons- inability reach, no longer interested, referral, transferred out of program** | **Few wanted medication alongside intervention NSHW perceived as caring Intervention was "very helpful." Impact- improved basic helping skills and problem-solving skills. 42% recommended more and/or longer sessions.** | **Participants said they would not change anything about where intervention took place or by whom it was delivered** | N/A | N/A | N/A | N/A | Sustainability of treatment effects especially in low- income women and mothers of young children. |

| Study ID | 1. Feasibility | 2. Acceptability | 3. Appropriateness | 4. Adoption | 5. Fidelity | 6. Cost | 7. Penetration | 8. Sustainability |
|---|---|---|---|---|---|---|---|---|
| Notiar 2021 | Screening tool- PHQ–9 17/19 agreed to participate 59% completed 12 session treatment Reasons for nonadherence- full time jobs, large groups | Satisfaction with the support. It helped gain knowledge, wellbeing and coping. Willing to recommend the support | All families allowed them to attend the intervention, and most of them attended all sessions. | N/A | N/A | N/A | N/A | N/A |
| Layton 2020 | All six NSHWs completed training and volunteered to participate in the study | Training was adequate, helpful and wholesome. Placements helped to practice skills. Supervision enhanced understanding Learning embedded in daily lives and led to personal development | N/A | N/A | N/A | N/A | N/A | Fear of intervention ending Feelings of concern and anxiety leading up to the completion of the support visits (n = 4) |
| Carter 2020 | Screening tool- Whooley questionnaire | NSHWs were appreciated for nonjudgmental attitude, time and empathy. NSHWs gratified to deliver intervention but dissatisfied over brief intervention. | N/A | N/A | N/A | N/A | N/A | N/A |
| Sikander 2019 | Screening tool- PHQ–9 78% completion rate Adherence- mean of 3·7/5 antenatal and a mean of 7·3/9 postnatal sessions completed 80% response rates | **Elevated social status as NSHWs** | **Family member engagement led to higher buy-in from family. Incentives motivated and sustained NSHWs** | N/A | Method: Observation Group classroom and field supervision Raters: Trainers Result: N/A | Mean cost: The THPP intervention cost US$133 per participant to deliver. | N/A | N/A |
| Horowitz 2013 | Screening tool- EPDS | Intervention as a source of support; Video feedback made aware and reflect on actions. | N/A | N/A | N/A | N/A | N/A | N/A |
| Appleby 2003 | N/A | Training was practical, useful and relevant leading to changes in knowledge, skills and competence post training | N/A | N/A | N/A | Pre training cost GBP 81, 116, 107 and post-training cost GBP 79, 108, 109 per patient, per depressed patient and per treated patient | N/A | N/A |
| Rahman 2008 | Screening tool- HRSD Recruitment method- Official register Drop out conditions- abortion, still born, premature or congenitally disabled born, infant death, given up for abortion, seriously ill mothers, migration. | N/A | N/A | N/A | N/A | N/A | N/A | N/A |

| Study ID | 1. Feasibility | 2. Acceptability | 3. Appropriateness | 4. Adoption | 5. Fidelity | 6. Cost | 7. Penetration | 8. Sustainability |
|---|---|---|---|---|---|---|---|---|
| Gureje 2019 | Screening tool- EPDS Lost to follow-up reasons- relocation or no contact 61% treatment completion rate | N/A | N/A | N/A | Method: Activity log (Clinical records) and direct observation Raters: Senior researchers Result: 58% were rated very good, 32% good and 10% poor | Mean cost: USD 46.85 per participant over 1 year in intervention arm vs. USD 24 in control | N/A | N/A |
| Prendergast 2001 | Screening tool- EPDS All completed 6 sessions. Dropouts were higher in control group | 70% of women in the intervention arm felt that the improvement was because of their NSHWs. | Health facility-based sessions (control arm) faced challenges relating to convenience, time arrangement and lack of concern on the mother. | N/A | Method: Audio recording Raters: Senior member (registrar) and psychiatrist Result: 70% of adhered to the protocol (problem-solving techniques and pleasant-event scheduling). | N/A | N/A | N/A |
| Dennis 2020 | Two step screening process with EPDS followed by Structured Clinical Interview for DSM-IV (SCID-I) Recruitment rate– 93.8% Response rate– 81.7% Treatment adherence– 86.7% Reasons for nonadherence listed | Facilitators- Convenience, competent NSHWs, satisfaction with the sessions 58.2% wished for more sessions | N/A | N/A | Method: Checklist and activity logs; Audio recording Raters: IPT experts Result: High treatment fidelity (86%) | N/A | N/A | N/A |
| Yator 2021 | Screening tool- EPDS | N/A | N/A | N/A | Method: Interpersonal inventory rating scale Raters: N/A Result: The mean score increased when IPT-G was offered to the waitlist control group. | N/A | N/A | N/A |
| Amani 2021 | Screening tool- EPDS Above 50% response rates 84% intervention adherence rates | N/A | N/A | N/A | N/A | N/A | N/A | N/A |

**Table 3.** (*Continued*)

| Study ID | 1. Feasibility | 2. Acceptability | 3. Appropriateness | 4. Adoption | 5. Fidelity | 6. Cost | 7. Penetration | 8. Sustainability |
|---|---|---|---|---|---|---|---|---|
| Rahman 2019 | 75% completed the assessment. | N/A | N/A | N/A | N/A | Conventional training cost- USD 170 per NSHW Technology assisted: USD 117 per NSHW | N/A | N/A |
| Atif 2019 | 75% NSHWs retained Dropout reasons- poor competency; migration; poor acceptance | Attendance in supervision sessions was above 85–100% throughout 3 years. | N/A | Linking to the health system led to greater buy-in from the community | N/A | N/A | N/A | N/A |
| Tomlinson 2020 | Screening tool- PHQ–9 Only 55% participated in 65% adherence rate. Reasons for missing sessions- child illness, household commitments, prohibition by family, dissatisfaction | N/A | N/A | N/A | N/A | N/A | N/A | 31/ 45 peer volunteers retained over 5 years. |
| Atif 2019 | N/A | Motivators – altruism, social network expansion, personal growth gained respect and strong supervision | Supervision appropriate to prepare for helping role | Logistical and coordination challenge to conduct group session | N/A | N/A | N/A | N/A |
| Ng'oma 2019 | Screening tool- EPDS | N/A | N/A | Barriers- busy and full clinics; no dedicated person, no guidelines. | N/A | N/A | N/A | N/A |
| Atif 2016 | Screening tool- PHQ–9 | Intervention was helpful which led to acceptability and motivation for NSHW. | N/A | N/A | N/A | N/A | N/A | N/A |
| Dennis 2010 | Screening tool- EPDS 63.3% response completion | Intervention alleviated stress, promoted coping, social integration Satisfied with the support, Found convenient, accessible Strong positive bonding with the NSHWs. | N/A | N/A | N/A | N/A | N/A | N/A |
| VanLieshout 2022 | Screening tool- EPDS Recruitment through social media, referral from the health facility. 33% lost to follow-up | N/A | N/A | N/A | N/A | N/A | N/A | N/A |

Note: Texts written in **bold** were extracted from secondary article.

Abbreviations: BDI – Beck's Depression Inventory; CES-D – Center for Epidemiologic Studies Depression; EPDS – Edinburgh Postnatal Depression Scale; IPT – Interpersonal Therapy; N/A – Not available; NSHW – Nonspecialist health worker; PHQ-9 – Patient Health Questionnaire-9; THP – Thinking Healthy Program.

Sawyer et al., 2019) were obtained from secondary publications (Segre et al., 2015). Training duration ranged from 4 h to 2weeks, some with follow-up sessions and refresher training. Lectures, audiovisuals and discussions were common methods used for theoretical content delivery, alongside role play, session observation (Dennis et al., 2020; Layton et al., 2020) or internships (Chibanda et al., 2014; Atif, Bibi, et al., 2019; Fuhr et al., 2019) to enhance skills. The use of technology such as telephones and tablets for training was also described in some studies (Dennis, 2003, 2010, 2013, 2014; Rahman et al., 2019; Nisar et al., 2020; Nisar et al., 2022). Training content focused on the assessment and treatment of mental health conditions based on a structured manual/protocol and was usually delivered by psychiatrists, psychologists or specialists.

### Supervision

The majority of the studies (n = 35) reported on supervision, with details of two studies retrieved (Brock et al., 2017; Leocata, Kleinman, et al., 2021) from secondary publications (Singla et al., 2014; Segre et al., 2015; Atif et al., 2017). Supervision primarily occurred face-to-face in-group settings on a weekly (n = 10), fortnightly (n = 1) or monthly (n = 11) basis or by need (n = 3) (Craig et al., 2005; Van Lieshout et al., 2020; Ransing et al., 2021). Electronic mediums such as telephones (Morrell et al., 2009; Dennis, 2013, 2014; Posmontier et al., 2016; Dennis et al., 2020), emails (Dennis, 2014) and apps (Eappen et al., 2018; Atif, Nisar, et al., 2019; Rahman et al., 2019; Yator et al., 2021) were also utilized. Supervision details (duration, frequency or content) were missing in nine studies (Slade et al., 2010; Letourneau et al., 2011; Carter et al., 2020; Layton et al., 2020; Leocata, Kleinman, et al., 2021; Leocata, Kaiser, et al., 2021; Notiar et al., 2021; Singla, MacKinnon, et al., 2021; Van Lieshout et al., 2022). Supervisors were predominantly mental health professionals, although peer-led supervision was common in studies involving peers as NSHWs. Some studies (n = 6) adopted a cascade model, where experts supervised local trainers who then supervised implementers (Atif et al., 2016; Atif, Bibi, et al., 2019; Atif, Nisar, et al., 2019; Rahman et al., 2019; Sikander et al., 2019; Leocata, Kleinman, et al., 2021). Supervision sessions mainly focused on reviewing intervention content, followed by practice sessions through role play, discussion on challenges faced during service delivery and potential strategies to manage burnout.

### Implementation outcomes based on Proctor's framework

An overview of the outcomes is provided in Table 3.

### Feasibility of interventions

Proctor's framework defines feasibility in terms of recruitment, retention and adherence to treatment. Altogether, 32 studies reported feasibility outcomes. Additionally, seven secondary articles were reviewed to extract data on feasibility.

**Recruitment and retention of service users.** Recruitment of perinatal women primarily occurred at the health facility, but social media and advertisements were also utilized. Out of 43 studies that reported screening tools, Edinburgh Postnatal Depression Scale (EPDS) was the most common (n = 24), followed by the Patient Health Questionnaire (PHQ-9) (n = 10), Center for Epidemiological Studies Depression (n = 2), Hamilton Rating Scale for Depression (n = 2), Beck's Depression Inventory (n = 2), Hospital Anxiety and Depression Scale (n = 1), Self-Reporting Questionnaire (n = 1) and Whooley's questionnaire (n = 1). Studies were able to recruit between 67 and 94% of the total eligible women. A

secondary article reported the lowest recruitment rate of 19% and cited language barriers, presence of comorbid conditions and experience of pregnancy loss as reasons for poor recruitment (Lund et al., 2020). Strict inclusion criteria often made recruitment a challenging and slow process (Letourneau et al., 2011), which was further exacerbated by unprecedented events such as COVID-19 (Amani et al., 2021).

Studies collecting data at multiple time-points generally had a 15–38% dropout at end line, but in some cases, dropout was as high as 91% (Husain et al., 2021). Retention was especially poor in studies that extended over 6 months in duration and studies involving urban minority low-income population (Zayas et al., 2004; Sawyer et al., 2019). Common reasons for poor retention were contact loss, hospitalization, time/interest constraints and program discontinuation.

**Recruitment and retention of service providers.** Five included articles (Roman et al., 2009; Atif, Nisar, et al., 2019; Van Lieshout et al., 2020; Nakku et al., 2021; Ransing et al., 2021) and one secondary article (Atif et al., 2017) reported the feasibility of recruiting, training and retaining the NSHWs. The feasibility of training and retaining NSHWs ranged from 67% to 100% (Roman et al., 2009; Van Lieshout et al., 2020; Nakku et al., 2021; Nisar et al., 2022). Common challenges pertaining to the retention of NSHWs included workload, transfer to different health facility, poor competency, migration, personal circumstances and poor acceptance by service users.

**Service users' adherence to treatment.** The treatment completion rate ranged from 31 to 100%. A study conducted in Afghanistan had the lowest treatment participation and retention, citing household commitments, refusal from family, dissatisfaction and unavailability of health staff (Tomlinson et al., 2020). An individual-focused intervention that had six sessions delivered at home had a treatment completion rate as high as 100% (Prendergast and Austin, 2001). Adherence was higher (95%) in a health facility-based intervention when embedded within regular postnatal visits (Chibanda et al., 2014). For a telephone-based intervention, the treatment completion rate was as high as 98% once the treatment was initiated (Dennis et al., 2020). This was the opposite for an app-based intervention where the user engagement reduced over time (from 64% to 14% over 16 weeks) (Sawyer et al., 2019).

Postnatal sessions were frequently missed, partly due to the tradition of mothers returning to their maternal home for postnatal recovery. As this often involved relocation, home-based sessions became logistically challenging (Leocata, Kleinman, et al., 2021). One secondary article found as low as 28% attendance in postnatal sessions (Lund et al., 2020). Sickness, experiencing loss, lack of time, stigma, fear of breaking confidentiality and dissatisfaction with the services or NSHWs were cited as reasons for not engaging in care.

### Acceptability

Twenty-four reviewed studies and seven secondary studies reported on acceptability.

**Service providers.** Self-driven, empathic and competent NSHWs were identified as key drivers to the intervention's success. NSHWs delivering interventions in person or electronically reported positive experiences, viewing the intervention delivery as an opportunity to serve others and expand their social network (Singla et al., 2020). They also perceived that the training and intervention delivery experience enhanced their knowledge, skills and confidence, contributing to personal development (Appleby et al.,

2003; Dennis, 2013; Glavin et al., 2010b; Layton et al., 2020; Boisits et al., 2021; Kukreti et al., 2022). Group supervision and tailored feedback helped address challenges and build confidence. Peer supervision, although beneficial, was less effective than expert supervision (Singla et al., 2020). Overall, NSHWs expressed satisfaction and willingness to engage in the future. Lack of confidence (Munodawafa et al., 2017; Carter et al., 2020), emotional burden (Dennis, 2013; Munodawafa et al., 2017) and resistance from family (Atif et al., 2016) hindered implementation.

Culturally appropriate content, illustrations and scripted guides better enabled NSHWs to deliver sessions (Dennis, 2014; Boisits et al., 2021; Leocata, Kaiser, et al., 2021). However, one study found that the violence-focused content was only beneficial for a specific demographic, suggesting its potential unsuitability as a universal intervention component (Ransing et al., 2021).

Service users. Engaging in the intervention yielded both physical and emotional benefits in service users. They expressed satisfaction with the NSHWs assigned to them. Educated, middle-aged females sharing similar language and culture were mostly preferred as NSHWs (Zayas et al., 2004; Singla et al., 2014; Nyatsanza et al., 2016). A strong match with the NSHW led to higher receptivity, trust and a strong bond (Dennis, 2010; Carter et al., 2020). Nurses were perceived as competent by 99% of the service users in one study (Dennis et al., 2020). However, NSHWs who read from manuals instead of engaging, did not give time, made invalidating remarks or set unrealistic hopes were seen as unhelpful (Slade et al., 2010; Davies et al., 2022).

Community-based health facility interventions were acceptable, and the provision of childcare eased attendance (Van Lieshout et al., 2020). Challenges included ill-equipped facilities and long waiting hours (Nyatsanza et al., 2016). Telephone-based support was accessible and alleviated concerns about transportation, time and childcare (Ross et al., 2013; Posmontier et al., 2016). For a mobile app-based intervention, a chat page where participants could communicate with NSHWs was the most used feature compared to a mood tracker or video content (Sawyer et al., 2019).

## Appropriateness
Small group training with a mix of classroom-based and practical sessions was perceived as most beneficial by the NSHWs (Layton et al., 2020). Both electronic-based and in-person training were deemed useful (Rahman et al., 2019; Nisar et al., 2022). NSHWs felt that these trainings enhanced their knowledge, confidence and readiness for their role (Dennis, 2014; Russell et al., 2020; Yator et al., 2021).

Interventions complementing the existing system and tailored to contextual issues were deemed more appropriate (Nyatsanza et al., 2016; Ransing et al., 2021). NSHWs reported difficulty with issues outside the intervention's focus (Munodawafa et al., 2017; Leocata, Kaiser, et al., 2021). Scripts provided structure, but some NSHWs found them constraining, highlighting a need for flexibility. Individual sessions allowed for discussing personal concerns and receiving tailored support (Slade et al., 2010), while group sessions fostered connections and normalized problems (Rahman, 2007; Russell et al., 2020; Van Lieshout et al., 2020). Service users preferred small groups and hesitated to engage in larger groups (Notiar et al., 2021).

Due to safety concerns and family resistance, home visits were less preferred by NSHWs (Zayas et al., 2004; Nyatsanza et al., 2016; Munodawafa et al., 2017; Leocata, Kleinman, et al., 2021). Phone- and app-based interventions were considered useful, user-friendly

and less stigmatizing, but women reported discomfort receiving calls in others' presence and missing each other's calls (Dennis, 2010; Ross et al., 2013). The chat function in apps was particularly useful for asking questions (Sawyer et al., 2019).

For peers, incentives in the forms of financial payments, transportation and communication compensation, gifts or household items were cited as one of the key motivators for engaging in service delivery (Atif, Bibi, et al., 2019; Ng'oma et al., 2019; Sikander et al., 2019; Leocata, Kleinman, et al., 2021).

## Programmatic adoption
Only nine studies reported on programmatic adoption. Brief interventions were easier to integrate into routine service at the health facility (Eappen et al., 2018; Boisits et al., 2021). Intervention delivery was easier for NSHWs when they linked their affiliation with the health facility (Atif, Nisar, et al., 2019). Health facility-based interventions had smooth functioning only when the health workers were cooperative. However, this placed an additional burden on NSHWs, requiring them to manage logistical, administrative and coordination tasks alongside providing psychological support (Zayas et al., 2004; Munodawafa et al., 2017; Atif, Bibi, et al., 2019). Lack of support from health facility staff (Zayas et al., 2004; Munodawafa et al., 2017), unequipped and inaccessible health facilities (Zayas et al., 2004; Eappen et al., 2018; Atif, Nisar, et al., 2019; Yator et al., 2021), lack of compensation and work burden (Atif, Bibi, et al., 2019; Atif, Nisar, et al., 2019; Ng'oma et al., 2019; Singla et al., 2020; Yator et al., 2021) hindered the implementation and adoption of the intervention in routine care. On the other hand, developing a maternal mental health guideline and creating a dedicated position within the health system were identified as facilitators for the integration of maternal mental health intervention into the health system (Ng'oma et al., 2019).

## Fidelity
A total of 18 studies reported on fidelity. Fidelity was assessed through the rating of session observations or audio recordings, or activity logs using guidelines, checklists or tools such as the Therapeutic Quality Scale (Fuhr et al., 2019; Sikander et al., 2019; Singla et al., 2020; Leocata, Kaiser, et al., 2021) and Interpersonal Inventory Rating Scale (Yator et al., 2021). Fidelity assessments were mainly done to ensure adherence to the study protocol (Fuhr et al., 2019), intervention content, use of clinical skills (Prendergast and Austin, 2001; Nyatsanza et al., 2016; Munodawafa et al., 2017; Fuhr et al., 2019; Gureje et al., 2019) and to identify challenges leading to targeted training/supervision (Rahman, 2007; Munodawafa et al., 2017). Higher scores in these assessments meant higher fidelity to the intervention, while lower scores generally indicated a lack of competency to provide care. While many studies reported that NSHWs had good adherence to the intervention (Prendergast and Austin, 2001; Slade et al., 2010; Munodawafa et al., 2017; Gureje et al., 2019; Dennis et al., 2020), four studies reported challenges such as NSHWs lacking effective communication skills and struggling to adequately explain the intervention component or follow the manual (Eappen et al., 2018; Layton et al., 2020; Boisits et al., 2021; Davies et al., 2022).

## Implementation cost
Altogether, five studies in the review reported cost analyses, of which three focused on the cost-effectiveness of the psychological intervention, whereas the other two focused on the training of NSHWs. Two studies reporting on the cost-effectiveness of the THP intervention in Pakistan and India reported that the

intervention was highly cost-effective, with an estimation of $1 per beneficiary (Fuhr et al., 2019) and each unit of improvement on the PHQ-9 score costing between $2 and 20 (Sikander et al., 2019). Another study in Nigeria comparing high-intensity over low-intensity treatment found no difference in terms of cost effectiveness (Gureje et al., 2019). A study in the United Kingdom found that training NSHWs improved their skills and led to positive changes in their clinical practices without increasing the overall cost of service delivery (Appleby et al., 2003). Another study comparing the cost of technology-assisted training against in-person training found technology-assisted training more cost-effective by 30% (Rahman et al., 2019).

### Penetration

Proctor's framework defines penetration as a level of institutionalization and maintenance of treatment at the systems level, usually occurring in the mid to late stages of implementation. This information was missing in the reviewed studies.

### Sustainability

Sustainability as institutionalization of treatment was not reported in the reviewed studies; however, four studies briefly outlined sustainability concerns. For example, engaging in short-lived projects affected NSHWs' motivation to engage fully (Atif, Bibi, et al., 2019). Service users and their families expressed similar worries (Ross et al., 2013; Atif et al., 2016; Nyatsanza et al., 2016). One study reported treatment effects after 8 weeks (Brock et al., 2017), while another reported retention of peer volunteers (68.88%) over 5 years, suggesting the sustainability of local NSHWs (Atif, Bibi, et al., 2019).

## Discussion

There is a growing need for more evidence in implementation science, which focuses on translating theories into practice, identifying facilitators and barriers and developing strategies to overcome challenges (Rapport et al., 2018; Bauer and Kirchner, 2020). Qualitative insights to document the implementation process are essential, as they can serve as a guideline to practitioners aiming to integrate perinatal mental health in their programs. We applied Proctor's framework of implementation science, which outlines implementation constructs and analyzes outcomes in the early, mid and late stages (Proctor et al., 2011), to report our findings. Our review found that most studies reported feasibility, acceptability, appropriateness and fidelity outcomes; however, very few evaluated cost, sustainability, adoption and penetration.

Our review indicates that acceptance and adherence were higher for interventions delivered at home or integrated in routine care when the NSHWs had matching characteristics with the service users. A strong bond with NSHWs was crucial, and without it, led to dissatisfaction with the program (Slade et al., 2010). For NSHWs, receiving training and supervision was a capacity-building opportunity, which enhanced their knowledge, confidence and readiness for the helping role (Dennis, 2014; Russell et al., 2020). None of the NSHWs had prior experience in mental health, therefore indicating the need for intensive training and supervision to maintain competency, ensure treatment quality, maintain fidelity and address emotional burnout (Watts et al., 2021). Incompetency of service providers can cause unintended harm (Dennis, 2010); hence, some studies in our review assessed competency when recruiting the NSHWs (Letourneau et al., 2011; Dennis, 2013, 2014; Munodawafa

et al., 2017; Fuhr et al., 2019; Dennis et al., 2020; Singla et al., 2020; Singla, MacKinnon, et al., 2021). Evidence also highlights the need for competency-based training in mental health to ensure quality and safety of the treatment (Kohrt et al., 2020). A cross-country study in LMICs on Ensuring Quality in Psychological Support (EQUIP), an online platform to assess competency, found that competency-based training was helpful in reducing harmful behaviors and improving helpful behaviors of the NSHWs (Pedersen et al., 2023). Breuer et al. (2018) found that regular supervision motivated the NSHWs to proactively screen and manage mental health problems. While supervision dosage can vary, quality supervision is arguably more important than the quantity of supervision (Kemp et al., 2019).

Even when interventions are feasible, acceptable and effective, their adoption in the health system cannot be guaranteed (Bauer and Kirchner, 2020). Very few studies in this review reported on systems-level implementation outcomes, such as adoption or sustainability, and none reported on penetration. NSHWs were often in a voluntary position and were trained to integrate psychosocial intervention into their regular work. While there was a good receptivity of the intervention by the NSHWs, they expressed being demotivated and overburdened without incentives. Further, the temporary nature of these interventions raised concerns about their sustainability, often affecting the motivation of both the service providers and service users to engage in the intervention (Ross et al., 2013; Atif et al., 2016; Nyatsanza et al., 2016; Atif, Bibi, et al., 2019).

Poor adoption and sustainability of evidence-based treatments pose significant challenges to address maternal mental health (Bauer and Kirchner, 2020). While Proctor's framework situates sustainability in the later implementation stages, emerging discourse suggests it is a continuous process spanning pre-, during and post-implementation phases (Pluye et al., 2004; Bergmark et al., 2018; Shelton et al., 2018). Program designers should proactively incorporate sustainability elements from inception, potentially through continuous stakeholder engagement to foster buy-in and cultivate an environment conducive to implementation. Strategies outlined by Vax et al. (2021) offer valuable guidance for implementing interventions.

Scaccia et al. (2015) emphasize the importance of assessing and ensuring "organizational readiness," defined as having the willingness and capacity to implement the innovation for adoption and sustainability. Innovation that fits well with the needs, culture, context and capacity of the organization is more likely to be adopted (Scaccia et al., 2015; Vax et al., 2021). However, the pervasive stigma associated with mental health poses a threat to adoption. Structural stigma, marked by inadequate policies, political will and investment, limits service availability, (Jenkins et al., 2013; Livingston, 2020) while community-level stigma delays help-seeking and reduces service utilization (Livingston, 2020). Addressing stigma therefore requires innovative strategies at multiple levels. At the systems level, careful planning, funding and evidence-based advocacy – supported by cost-effectiveness studies – are essential for political buy-in and institutionalization within the health system (Bergmark et al., 2018; Vax et al., 2021). Meanwhile, at the community level, sensitization and engagement activities can foster awareness and encourage service uptake (Ng'oma et al., 2019; Subba et al., 2024).

The WHO's guide for integration of perinatal mental health in maternal and child health services provides practical guidance for planners and policymakers on "what" actions can be taken to embed these interventions into routine care (WHO, 2022). However, a deeper understanding of "how" to implement these

interventions in real-world settings and "what works and what does not" (including facilitators and barriers) remains essential. Despite the growing prominence of implementation science, the paucity of studies reporting process evaluation and implementation outcomes for perinatal depression interventions hinders the identification, replication and synthesis of evidence. Future studies could address this gap by using frameworks such as the Standard for Reporting Implementation Studies to report their findings, making them more visible and accessible (Pinnock et al., 2017).

## Limitations

The limitations of this review include the exclusion of non-English publications, which might have resulted in the omission of relevant articles. Second, this study conducted a narrative synthesis of the implementation constructs. For implementation constructs such as feasibility and fidelity that predominantly use quantitative measures, future research could consider conducting statistical analyses. Third, this review only focused on treatment interventions delivered by the NSHWs to the adult population. Given the wide engagement of NSHWs in prevention and promotion interventions globally and the focus across all age groups, this review could have excluded some important studies involving perinatal adolescents and girls.

## Conclusion

This review synthesized evidence on implementation outcomes using Proctor's framework to gain insights into the process, success and barriers of NSHW-delivered psychosocial interventions. Findings indicate that such interventions are well-accepted, and NSHWs can effectively deliver them when adequately trained, supervised and incentivized. However, there is a notable lack of studies exploring systemic factors influencing adoption, maintenance and sustainability. Further research is needed to elucidate the factors affecting the systems level integration of these interventions. Future implementers would benefit from employing implementation science frameworks to guide planning, execution and sustainability while considering various implementation factors across different stages.

## Abbreviations

| | |
|---|---|
| CBT | cognitive behavioral therapy |
| HIC | high-income countries |
| HIT | high-intensity treatment |
| IPT | interpersonal therapy |
| LIT | low-intensity treatment |
| LMIC | low- and middle-income countries |
| NSHW | nonspecialist health workers |
| PRISMA | preferred reporting items for systematic review |
| WHO | World Health Organization |

**Open peer review.** To view the open peer review materials for this article, please visit http://doi.org/10.1017/gmh.2025.10010.

**Supplementary material.** The supplementary material for this article can be found at http://doi.org/10.1017/gmh.2025.10010.

**Data availability statement.** The authors confirm that the data supporting the findings of this study are available within the article, references and/or its supplementary files.

**Acknowledgments.** The authors would like to thank the ENHANCE Collaborative Learning Group members for their continuous support and feedback. The authors would also like to thank Alexandra Blackwell for her valuable assistance with language editing.

**Author contribution.** SS, AR, NPL and PS1 designed the study, and drafted the protocol. SS and PS1 developed the search strategy. PS1 and PS2 searched, screened title and abstracts, retrieved full texts and performed data extraction. PS1 performed quality assessment of included studies, synthesized data and drafted the manuscript under close supervision from AW. PS2 drafted the methods section. AR, NPL and AW provided initial feedback to the manuscript. All authors reviewed the final version of the manuscript and gave approval for submission.

**Financial support.** This study was supported under financial aid from the National Institute for Health and Care Research (NIHR), UK's RIGHT CALL 2 NIHR200817 ENHANCE: Scaling-up Care for Perinatal Depression through Technological Enhancements to the 'Thinking Healthy Program' (RIGHT CALL 2 NIHR200817). Further information is available at https://fundinga wards.nihr.ac.uk/award/NIHR200817. The funding agency has no role in the collection, management, analysis and interpretation of data or the decision to submit the report of publication.

**Competing interests.** The authors declare none.

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
