## [Reviewer Report]

Thank you for the opportunity to review this paper which I found fascinating.

This systematic review is a timely and comprehensive synthesis of the implementation science evidence for psychological interventions for perinatal depression delivered by non-specialist health workers (NSHWs). The authors apply Proctor’s implementation science framework, which offers valuable real-world insights into the pragmatic and logistical processes associated with these interventions. In low-resource settings, the increasing focus on task-shifting or sharing in mental healthcare, makes this review an important and novel contribution to the literature on how NSHWs can effectively address perinatal mental health needs. The findings highlight the feasibility and acceptability of these interventions when they are culturally adapted and integrated within existing healthcare systems. The vital role of training, supervision, and incentivization for NSHWs, is underscored with clear implications provided for policymakers and healthcare practitioners. The review enhances our understanding of how to design more effective and sustainable perinatal mental health programs, while identifying key areas for future research on system-level integration and long-term sustainability.

This is a global review which is successful in including evidence from a range of contexts, covering low, middle and high income settings. However, given the importance of task sharing for perinatal mental healthcare in resource constrained settings, the review could be strengthened in the analysis and discussion sections, by highlighting trends emerging and commonalities or differences between these settings.

Mostly minor comments and suggestions follow, according to the relevant section in the paper. They mainly relate to grammar and the need for improved clarity.

Introduction

• As the authors included anxiety in their search, some background on perinatal anxiety disorders should be included here.

• Line 32 – ‘population’ should be plural. Please check throughout the paper for correct use of the singular versus the plural case.

• Line 33 – missing ‘a’ before ‘rural’. Please check throughout the paper for frequently missing articles.

• The use of prepositions throughout the paper requires careful review. I have only highlighted some of these instances.

Search strategy

For the nonspecialist category of keywords, it is a pity that ‘midwives’ were not included. This cadre is not necessarily subsumed within nursing cadres in many countries’ health systems, and would be a relevant NSHW for perinatal mental healthcare. Thus, although studies involving midwives may largely have been included under the ‘nurses’ keyword, there may well have been studies using midwives as the NSHW which would not have been detected in your search.

Quality Assessment

The article ‘the’ before ‘scoring system’ in line 36 is confusing.

Data Synthesis

It is not clear what ‘secondary articles’ refer to, nor what ‘examined to verify data inclusion’ means.

Quality Assessment

This section requires a review of grammar: use of articles and commas etc. for the meaning to be made more clear. Line 22 – ‘were’ instead of ‘was’.

Table 2 – please clarify what ‘X’ means in the key. I assume it refers to no data being available for this category, but this should be made explicit.

Intervention details

For the three studies targeting ‘maternal depression’, how was this defined to be separated by you into a category distinct from perinatal or postnatal depression.

Please check grammar here.

Supervision – please check grammar in the last sentence.

Table 3 – there are several acronyms used that are not detailed in the key or in the list of abbreviations

Recruitment and Retention – Does ‘poor acceptance by women’ refer to women using the service?

Adherence to treatment

• I suggest a preposition change: embedded ‘within’ regular…

• Would a reason for non-engagement not rather be ‘the fear of breaking confidentiality’?

Service Users

• It is not clear what ‘assigned roles’ refers to. The term ‘participants’ is also confusing in this context. Are these service users’ preference for the type of provider?

• I suggest rephrasing the sentence starting ‘However’ line 53 for clarity and grammar.

Appropriateness

• I suggest you clarify in the first line of the first paragraph, that the training refers to NSHWs.

• The incentives paragraph is a bit confusing with respect to your reference to service users versus NSHWs.

• There seems to be a missing word or phrase at the end.

Adoption

• It does not make sense that extra logistical etc. burden were reported to gain health worker buy-in.

• The last sentence should be reworked for grammar and clarity.

Fidelity

• Should the ‘interpersonal inventory rating scale’ be capitalised?

• I suggest the term ‘more suggestive’ be changed.

• It would be interesting to include data on what was attributed, by study authors, to good versus poor fidelity.

Implementation Cost

I suggest this section is reworked for grammar and clarity.

What does ‘post training NSHWs skills’ mean?

Penetration

I suggest this run-on sentence be rewritten for clarity.

Sustainability

The term participants is a bit confusing and should be checked throughout. Research participants could be service users or NSHWs or others. Please specify where relevant.

In the last line, replace ‘suggested’ with ‘suggesting’.

Discussion

This section would benefit from substantial English editing.

It was not clear to me from your data that interventions delivered at home were more acceptable or were associated with higher levels of adherence.

Lack of NSHW competence was not adequately referred to in the results and thus the discussion element here seems a bit out of place. As this is highly relevant, perhaps the relevant section in the results could be expanded.

Line 26 – Does the engagement here refer to service user engagement with the intervention or NSHW engagement with the programme?

Line 36 – Does the receptivity refer to that of the users or the providers (or possibly, managers)?

Line 50 – Does the term ‘internalization’ refer to integration? If so, I suggest you consistently use the latter.

Line 56/57 – I suggest deleting ‘of innovations’.

Line 6 – For clarity and grammar, I suggest rewording the sentence starting with ‘Further, …’

I suggest giving the full name for the WHO guideline.

It is not clear why a new framework for reporting implementation studies is introduced at the end when the authors used Proctor’s framework.

Given your discussion on adoption, cost, penetration, readiness and sustainability, did none of the included articles refer to engagement with senior health officials and programmers? I am aware of several studies selected that did so.

Limitations

Given the likely suitability of NSHWs to provide preventive or promotive interventions, it is a pity these were not specifically included in the search. However, I acknowledge they may have been included in ‘multicomponent’ or ‘comprehensive psychosocial intervention’ etc. This may be worth a brief mention in limitations.

Conclusion

Line 42 - The paper previously refers to ‘sustainability’. I suggest this is used consistently here.

---

## [Reviewer Report]

Thank you for the opportunity to review this manuscript. The authors cover an important topic, and the use of Proctor’s Implementation Science Framework is a useful scaffold to frame the paper. The paper requires a thorough grammar edit (the use of singular/plural and punctuation, etc.). There are several points that should be addressed/clarified and defined. However, the results are clearly written and may be useful for programs that are interested in task shifting approaches to address perinatal mental health. Detailed comments below.

Introduction.

1. First paragraph refers to ‘debilitating effects on the woman – however there is no information provided on what these effects are. The readers will understand the importance of perinatal depression better if there is information on why it is important (what the effects are).

2. Row 7: Should read ‘marginalized populations’. Plural

3. Just an observation that the authors chose to use an example from rural USA – this is a very specific example and leads the reader to believe that this study will focus on perinatal depression in the USA given it is in the first paragraph of the manuscript. If this is a global study, suggest adding in an example that can be universally understood/applicable.

4. Last sentence of first paragraph: Would also add that before lack of available treatment, there is low provider awareness and lack of skills to train for perinatal depression. Also mental health stigma and provider discomfort further impede recognizing depression symptoms.

5. Before moving into task shifting in the second paragraph, there should be some discussion about the shortage of mental health care specialists. From my perspective, the treatment gap refers to the high prevalence of mental illness and the shortage of mental health care providers. There should be some discussion around this space.

6. Overall the introduction could be strengthened by more information on why PMH is important, who the most at-risk groups are, and why people in LMIC experience higher prevalence rates. Also of note, the authors refer to ‘perinatal women’, however perinatal adolescent girls face significantly higher rates of perinatal mental health symptoms. There should be some discussion around this – and/or suggest using the term ‘perinatal girls and women’.

7. The last paragraph of the introduction can be strengthened by information on how the information from this systematic review can be used and by whom.

Methods

8. Last sentence of the first paragraph is unclear. What is inadequate information? Typically within systematic reviews, if another review on the topic was found, the reference list of the review would be found. For an original study, it is not clear why a reference list would be searched if the study did not provide sufficient information.

9. Based upon the comment about adolescent girls above, why were only girls age 18 and above included?

10. It would be helpful to define the prevention interventions that were excluded. For example psychoeducation can be considered a prevention intervention, but it could also be considered a therapeutic intervention. Clear definitions will be helpful.

11. Small note, the reviewers initials are introduced here – though they are mentioned above. Suggest placing the initials where they are first mentioned in the methods under search strategy.

12. Data extraction – PICO is mentioned here – but it typically refers to inclusion criteria. Suggest providing more clarification for the inclusion criteria above through the PICO framework.

13. Data extraction: Is there any information available on study inclusion alignment (what percent of studies did the authors agree on initially, what % required further discussion._

14. From the methods it is also unclear if a traditional two-step process was used wherein the authors first conducted a TIAB review blindly – came together to make determinations, and then blindly did a full-text review and then joining for a similar final determination process.

Results

15. Overall the results read very well. A note that in several places the authors use the term ‘some’ and in other places quantify the number of interventions. Suggest quantifying in place of using the term ‘some in all instances.

16. Recruitment and retention: From the title it is unclear if the authors are referring to patients or to non-specialists. This should be clarified in the title or the first sentence.

17. Given the participants were recruited to enter a mental health intervention, one would consider a positive screen for mental health symptoms as the process of recruitment into a therapeutic intervention. Can the authors clarify?

18. The recruitment paragraph is also quite vague – why was recruitment slow and challenging? It seems that some generalizations (beyond COVID) could be made here.

19. Adherence to treatment: The sentence that begins with ‘in some cultures’ is unclear. How does the tradition of visiting maternal house for postnatal care reduce postnatal participation in a mental health intervention?

20. 3.5.2.1 Service providers: The sentence beginning with culturally appropriate: ‘facilitated’ does not seem like the correct term here. Suggest replacing it with ‘enabled’.

21. 3.5.2.2 Service users: The last sentence of the first paragraph seems to have a word missing. After ‘however’ it appears that there should be a noun – NSHWs?

22. How were ‘community health facility-based interventions defined? It seems like the community health should be removed here.

23. Table 3: Consider formatting this differently – As is, the studies are challenging to differentiate from one another.

Discussion

24. Overall the discussion summarizes the information well. The paragraph starting with Scaccia et al focuses on organizational readiness. However, in most LMIC, system readiness is equally an important factor for utilization of NSHW in perinatal depression interventions. For example, lack of supportive policies that enable task shifting, inadequate governance, etc. can create significant challenges. Suggest discussion in this realm.

25. Stigma is also a significant concern in many communities – the results indicate that peer-based NSHWs were preferred, but very often people will avoid accessing mental health services within one’s own community due to mental health stigma – the discussion could be enriched by bringing in additional literature on this topic given how pervasive the issue of stigma is globally.

---

## [Editor Report]

Dear Authors - 

Thank you for the opportunity to review this manuscript.

Please carefully consider and respond to the Reviewers' comments. 

Also requested is a careful grammar edit to improve readability.

---

## [Reviewer Report]

Thank you for working on the revised manuscript and for the opportunity to review this draft. The authors have been responsive regarding initial feedback and the information contained within the manuscript is interesting. There are, however, still some changes that can be made to strengthen clarity of writing and to further increase its utility for readers.

For further clarity in the first paragraph of the introduction – suggest breaking this out into 2 paragraphs. The first paragraph should end after the marginalized population description. Suggest the second paragraph then clarify that PMH service demand by clients is the first issue, followed by PMH service availability, and then the systems issues. The information that is already available within the manuscript can fit into these three buckets. This will help to provide further structure to the introduction in general.

Please remove quotes around “treatment gap” in the sentence: To address the “treatment gap” in mental health, the World Health Organization (WHO) advocates for a task-sharing approach – whereby expert knowledge and skills are transferred to non-specialist health...

Last paragraph of introduction: Suggest breaking out this sentence into two sentences or streamlining the sentence in some way, it is currently convoluted: As the NSHWs continue to be an important cadre for delivering services for perinatal depression, it is important to understand facilitators and barriers to their on how best they can be mobilization; to establish feasibility and acceptability and maintain fidelity and to draw lessons for the future.

Delivery agents and their characteristics: The sentence starting with ‘In Zimbabwe’: what does it mean to have experience with HIV/AIDS? Do the authors mean that the peers and participants had comorbid HIV and postnatal depression? Also – just check the journal guidelines – flagging terminology with HIV/AIDS. Is this the preferred term or is it just HIV?

Service users: The last sentence that was added in the second paragraph should be re-worded to clarify that common reasons for POOR retention were…

Service providers: For the reader, it would also be helpful to know what the common enablers to retention were.

Adherence to treatment: Overall, I find this paragraph confusing – the authors state that attendance was as low as 28% but then move directly into attribution to the tradition of visiting a maternal house for postnatal care – would not his increase attendance? The following sentence also has confusing language regarding the telephone-based intervention with 75% treatment initiation, BUT 98% completion. Why is there a ‘but’ used here? This may just be a language/editorial issue, but overall, I would encourage the authors to draw upon the skills of an editor as the information here seems to be valuable, but the message is getting lost. It also may be helpful to the reader to understand the duration of each intervention specifically highlighted as an example within this paragraph.

Appropriateness: The last sentence in this paragraph mentions incentives – but it is not clear who the incentives are for. Is this for the NSHWs to provide services, or for the clients to access services? More clarification in this sentence will be helpful.

Adoption: Given the focus of the manuscript is on PMH, suggest clarifying that the paragraph on adoption is programmatic adoption.

Implementation cost: Please check the references here – it appears that the reference next to Fuhr 2019 requires fixing.

Penetration: Please remove the term ‘however’ at the beginning of the second sentence. It is not necessary in this context.

Discussion: The first paragraph primarily focuses on the need for more evidence from implementation science – while this is true – it seems to step away from the focus of this manuscript which would be strengthened by providing a brief summary of fundings at the beginning of this section. The review also included 56 studies- which is a great deal in comparison to many systematic reviews, so the point that there is a dearth of evidence in this regard does not seem accurate.

The discussion as a whole lacks structure and could be further clarified by providing some sub-headings/a framework so that the authors’ interpretation of the results is more easily understood.

The paragraph beginning with ‘allocating’ seems to be free floating within the discussion. Where does this fit? It needs to be tied into the existing language.

Also suggest within the discussion that the systematic review did not examine associated outcomes, and then consider drawing in some of the evidence available that can further help the reader to understand why PMH is a priority.

---

## [Reviewer Report]

Second review

The paper has been substantially strengthened and will be an excellent and highly useful contribution to the literature.

I have some very minor revisions suggested.

Under 3.5.1.1.1 Service users – ‘Retention was especially poor in studies that extended over 6 months and for urban minority poor population (Sawyer et al., 2019; Zayas et al., 2004).’ – the several descriptors for populations (plural?) are confusing. Is this one category? Should the adjectives be separated by a comma?

‘Common reasons for retention were contact loss, hospitalization, time/interest constraints, and program discontinuation,’ – these seem to be reasons for non-retention. There is a missing full stop at the end.

The term ‘maternal house’ or ‘maternal home’ may not be well understood in many contexts. I suggest some clarification.

In the table 3, I suggest the term ‘contactless’ be changed to ‘uncontactable’

Limitations

I am not clear why the lack of inclusion of studies involving children would have been relevant for perinatal CMD interventions. Are you referring to perinatal adolescents and girls?

---

## [Editor Report]

Dear Authors,

Thank you for your re-submission of this manuscript in response to the Reviewers. 

The Reviewers have some remaining questions and concerns that I hope you’ll be willing to address.

---

## [Reviewer Report]

Thank you for the revision. The manuscript is much improved. Small suggestions:One final suggestion. Please change 3.5.4 Adoption to ‘Programmatic Adoption’. The section heading is confusing in this context without clarifying what adoption refers to.

---

## [Reviewer Report]

Thank you for the opportunity to review this manuscript again. This version has satisfactorily taken into account review comments.

Kindly note minor suggestions remaining for which I do not need to review again:

1. In the impact statement - the word ‘are’ should be removed; the last phrase ‘at system level’ could be rephrased to be more clear.

2. The response to reviewer 1 on the discussion section pertaining to stigma: the response letter includes a restructured paragraph that reads well. However, the first phrase of this paragraph is not worded the same in the amended manuscript (tracked version). I suggest using the response letter version.

3. The response to reviewer 2 regarding the limitations section does not appear in the amended manuscript (tracked version).

---

## [Editor Report]

Dear Authors,

Thank you for your revised manuscript. The Reviewers recommend acceptance of your manuscript for publication pending a few remaining minor issues. 

In addition, please carefully proofread your revised manuscript prior to resubmission, as I noticed a few grammatical issues (examples: Impact statement, Lines 2-3, I believe the word “that” is missing after the word “anxiety”; Introduction, lines 33-34, missing word “base” after “evidence”; Introduction line 47, I believe should read, “constructs is [not ”are“] still lacking” (referring to a singular “global evidence on evalution”).

Thank you for your attention to these minor details, and I look forward to your revised manuscript.